# Antioxidative Effects of *Thymus quinquecostatus* CELAK through Mitochondrial Biogenesis Improvement in RAW 264.7 Macrophages

**DOI:** 10.3390/antiox9060548

**Published:** 2020-06-23

**Authors:** Jin Young Hong, Hyunseong Kim, Wan-Jin Jeon, Seungho Baek, In-Hyuk Ha

**Affiliations:** 1Jaseng Spine and Joint Research Institute, Jaseng Medical Foundation, Seoul 135-896, Korea; vrt23@jaseng.org (J.Y.H.); biology@jaseng.org (H.K.); cool2305@jaseng.org (W.-J.J.); 2College of Korean Medicine, Dongguk University, 32 Dongguk-ro, Ilsandong-gu, Goyang-si, Gyeonggi-do 10326, Korea; Baekone99@gmail.com

**Keywords:** mitochondria, reactive oxygen species, oxidative stress, macrophage

## Abstract

Oxidative stress plays a key role in the pathogenesis of several diseases, including neurodegenerative diseases. Recent studies have reported that mitochondrial dysfunction is a leading cause of the overproduction of reactive oxygen species and oxidative stress. Mitochondrial changes play an important role in preventing oxidative stress. However, there is a lack of experimental evidence supporting this hypothesis. *Thymus quinquecostatus* CELAK (TQC) extract is a plant from China belonging to the thymus species, which can mediate the inflammatory response and prevent cell damage through its antioxidant activities. This study examines whether TQC can scavenge excess ROS originating from the mitochondria in RAW 264.7 macrophages. We used lipopolysaccharide (LPS) to induce inflammation and oxidative stress in RAW 264.7 macrophages and performed an immunocytochemistry dot blot of 8-hydroxy-2′-deoxyguanosine (8-OHdG) and real-time PCR to analyze the expression levels of genes involved in mitochondrial biogenesis and oxidative metabolism. TQC was found to significantly reduce the intensity of immunostained MitoSOX and 8-OHdG levels in the total genomic DNA within the mitochondria in RAW 264.7 macrophages. The HO-1 and Nrf2 mRNA levels were also significantly increased in the TQC groups. Therefore, we verified that TQC improves mitochondrial function and attenuates oxidative stress induced by LPS. Our results can provide reference for the effect of TQC to develop new therapeutic strategies for various diseases.

## 1. Introduction

An excessive increase in intracellular reactive oxygen species creates oxidative stress and prevents cells from functioning normally, ultimately inducing cell death. Oxidative stress has been suggested to be an important pathophysiological mechanism in a number of neurological diseases, in addition to cardiovascular and degenerative diseases; therefore, the importance of ROS has been demonstrated [1,2,3]. Mitochondria are the key organelles that produce ROS, and, in normal mitochondria, 90–95% of the total oxygen is used to produce adenosine triphosphate (ATP), whereas 1–2% is converted to ROS during cellular metabolism [4].

These ROS, the generation of which is inevitable, play a role in the destruction of foreign invaders, such as bacteria or viruses, and in hormone regulation. However, when mitochondria are damaged and morphological and functional changes occur, most of the oxygen is used to produce ROS. The excessive production of ROS causes DNA damage and creates an oxidative stress environment that induces cell death and tissue damage [5,6]. Therefore, recent studies have suggested that the excessive release of ROS, and dysfunction due to morphological and functional changes in mitochondria, could be the main cause of oxidative environments. Furthermore, studies are underway to develop new therapeutic strategies of mitochondrial damage inhibition or the minimization of the excessive production of ROS by investigating related mechanisms [7,8,9].

New therapeutic strategies involve regulating mitochondrial function to control the oxidative stress environment. Furthermore, antioxidants and metabolic agents have been known to be associated with mitochondria, and their excellent antioxidant activities have improved their effects on mitochondrial functions [10,11]. For example, studies have reported that antioxidants extracted from food-derived natural substances and medicinal plants protect the body from various diseases by controlling damage to the mitochondrial DNA, thereby inhibiting the secretion of ROS [12].

*Thymus quinquecostatus* CELAK (TQC), an aromatic herb belonging to the family Lamiaceae, is distributed in South Korea, Japan, China, Mongolia, and India. TQC contains a considerably large number of flavonoids and phenolic compounds, and its excellent antioxidant activity has been demonstrated in numerous studies. However, studies on its basic mechanisms are lacking [13,14,15].

Many previous studies have provided clear evidence that the extracts pretreatment is important for improving extract efficiency and increasing their endogenous antioxidant activity against free radicals [16,17,18]. In the present study, we also investigated the antioxidative effects of TQC extract pretreatment that are associated with changes in mitochondrial function for 1 h before a 24 h treatment with lipopolysaccharide (LPS). We confirmed the morphological and functional changes in the mitochondria induced by lipopolysaccharide (LPS) in macrophages and demonstrated that TQC reduces oxidative stress by promoting the recovery of mitochondrial function and inhibiting the production of ROS. This is the first study to demonstrate the association between the antioxidant effects of TQC and mitochondrial function recovery and presents possible mechanisms of mitochondrial dysfunction, which is the underlying cause of various diseases, and a new therapeutic strategy.

## 2. Materials and Methods

### 2.1. Preparation of Extracts from the TQC

The TQC was prepared with water via refluxing for 3 h and cooled at −20 °C. It was filtered once with filter paper (Hyundai micro, HA-030, Seoul, Korea) at room temperature (RT). The filtrate was lyophilized by a freeze dryer (Ilshin BioBase, Gyeonggi-do, Korea) to obtain TQC dry extract. The dried extract was weighed, the extract yield was calculated, and the extract was dissolved in phosphate buffered saline (PBS) to the desired concentration. It was then moved to a conical flask before use and maintained at −20 °C.

### 2.2. Cell Culture and Treatment

RAW 264.7 macrophages were cultured in Dulbecco’s modified Eagle’s medium (DMEM) containing 10% heat-inactivated fetal bovine serum (FBS), streptomycin (100 µg/mL), and penicillin (100 U/mL) at 37 °C in an incubator under a humidified atmosphere of 5% CO_2_:95% air. Adherent cells were mechanically detached by a sterile cell scraper and plated onto 24, 48, or 98-well plates at 70–80% confluence. TQC was dissolved in PBS, and cells were pretreated for 1 h with TQC extract in a final concentration of 50, 100, and 200 µg/mL. After 1 h incubation of the cells with extracts, LPS was added to a final concentration of 1 μg/mL for 24 h. Samples were then divided into five groups (n = 6/group): Blank group; no-treatment group; Control group; LPS only treatment group; TQC_50 group; 50 µg/mL TQC pretreated + LPS group; TQC_100 group; 100 µg/mL TQC pretreated + LPS group; TQC_200 group; 200 µg/mL TQC pretreated + LPS group. We have outlined our experimental procedures in more detail in a timetable, which we have added to the methods section as Scheme 1.

### 2.3. Cell Proliferation Assay

The cells were pretreated for 1 h with TQC extract in a final concentration of 50, 100, and 200 µg/mL, followed by stimulation with or without LPS. Following a 24 h incubation period, the cellular proliferation was assessed using the EZ-CyTox cell viability assay kit (Daeillab, Seoul, Korea). EZ-CyTox solution (10 μL) was added to each cell-cultured 96-well plate, and the mixtures were incubated for 2 h at 37 °C. Absorbance was then measured using a microplate reader (Epoch, BioteK, Winooski, VT, USA) at 450 nm. Cellular proliferation was expressed as a percentage of the LPS-treated cells, which were defined as 100% viable.

### 2.4. Nitric Oxide and H_2_O_2_ Assay

The nitric oxide assay was performed to quantify NO production. Briefly, RAW 264.7 macrophages were pre-incubated with the TQC extract and LPS (1 µg/mL) for 24 h; NO production was measured using Griess reagent (2% sulfanilamide in 5% phosphoric acid, Sigma-Aldrich, St. Louis, MO, USA). Then, 50 µL cell culture medium was mixed with 50 µL Griess reagent at a ratio of 1:1 (*v*/*v*). Subsequently, the absorbance of the mixture was measured using a microplate reader at 540 nm. Fresh culture medium was used for the blank in all experiments. The quantity of nitrite was determined using a sodium nitrite standard curve. Finally, H_2_O_2_ production was detected by the OxyBlot oxidized protein detection kit (23280, Thermo Fisher Scientific, Waltham, MA, USA), according to manufacturer’s protocol.

### 2.5. Mitochondrial ATP Assay

ATP is produced in the mitochondria for energy supply. ATP production by the mitochondria is essential for cell respiration and metabolic homeostasis. ATP levels were measured using the CellTiter-Glo^®^ luminescent cell viability assay (Promega, Madison, WA, USA), according to the manufacturer’s instructions. All of the assays were performed in the opaque-walled 96-well plates with cells in culture medium. Briefly, the cells were prepared in opaque-walled 96-well plates at a density of 1.5 × 10^4^ cells/well and were pretreated with the TQC extract before being added to LPS in experimental wells for 24 h. A volume of CellTiter-Glo^®^ reagent, equal to the volume of cell culture medium present in each well, was added, and the contents were mixed for 2 min in an orbital shaker to induce cell lysis. The plate was incubated at RT for 10 min to stabilize the luminescent signal, and the luminescence was examined using the GloMax^®^ Navigator Microplate Luminometer (Promega). The ATP standard curve was prepared by a serial ten-fold dilution of ATP solutions in culture medium, to which a volume of CellTiter-Glo^®^ reagent, equal to the volume of ATP, was added in each well. The luminescence values were determined using a standard curve.

### 2.6. Immunocytochemistry (ICC)

Immunocytochemistry (ICC) was performed to analyze the antioxidant and mitochondrial changes in the LPS-activated RAW 264.7 macrophages. After 24 h, the samples were fixed with 4% paraformaldehyde for 30 min and rinsed three times for 5 min each with PBS. The cells were incubated with 0.2% Triton X-100 in 1× PBS solution for 5 min, rinsed twice with 1× PBS for 5 min, and blocked with 2% normal goat serum (NGS) in 1× PBS for 1 h. The primary antibodies were diluted in 2% NGS, and the slides were incubated overnight at 4 °C. The primary antibodies used were as follows: rabbit anti-inducible nitric oxide synthase (iNOS; 1:100, Abcam, Cambridge, UK), mouse anti-8-hydroxydeoxyguanosine (8-OHdG; 1:100, Santa Cruz, CA, USA), rabbit cytochrome c (CycC; 1:100, Cell Signaling Technology, Beverly, MA, USA), mouse anti-heme oxygenase 1 (HO-1; 1:100, Abcam), rabbit anti-BAX (1:100, Santa Cruz), and rabbit anti-Caspase 3 (Casp3, 1:100, Cell Signaling) were placed in 2% NGS and incubated for 2 h in fluorescent secondary antibodies (FITC-conjugated goat anti-rabbit IgG, 1:200; FITC-conjugated goat anti-mouse IgG, Jackson Immuno-Research Labs), diluted at 1:200 dilution in 2% NGS. The sections were washed three times for 5 min with PBS, mounted with fluorescence mounting medium (Dako Cytomation, Carpinteria, CA, USA), and imaged using confocal microscopy (Eclipse C2 Plus, Minato, Tokyo, Nikon, Japan). For the quantification of fluorescence intensity, five representative images were captured with 400× magnification, and all of the images were taken with fixed acquisition settings in confocal microscopy. The average intensity was measured using Image J software (1.37 v, National Institutes of Health, Bethesda, MD, USA).

### 2.7. Mitochondrial Staining

MitoSOX-based assays are widely used to quantify cellular ROS, especially mitochondrial superoxide in live cells. RAW 264.7 macrophages were seeded on glass coverslips in 24-well plates at a density of 5 × 10^4^ cells/well and cultured for 24 h. After incubation, the cells were washed twice with HBSS/Ca^++^/Mg^++^ (GIBCO). The cells were pre-incubated with 5 µM MitoSOX™ red mitochondrial superoxide indicator (Thermo Fisher Scientific) for 30 min at 37 °C in the dark. The cells were washed once, and the cell medium was changed. After 6 h, the cells were pretreated with TQC extract to give a final concentration of 50, 100, and 200 µg/mL. After a 1 h incubation of the cells with extract, LPS was added to a final concentration of 1 µg/mL for 24 h. The cells were fixed with 4% PFA for 5 min, permeabilized with 0.2% triton × 100 for 5 min, and treated with DAPI at a concentration of 1 µg/mL for 10 min. Stained samples were then mounted onto glass slides using Dako mounting medium. Additionally, we confirmed the morphological change of mitochondria using MitoTracker^®^ Red FM (Molecular Probes Inc., Eugene, OR, USA). The cells were plated on glass coverslips in 24-well plates at a density of 1 × 10^4^ cells/well. After the treatment of the extract and LPS, as previously described, the live cells were stained with pre-warmed staining solution containing 200 nM MitoTracker. The cells were incubated for 30 min under growth conditions in a CO_2_ incubator. After staining was complete, the staining solution was replaced with fresh pre-warmed medium, and the cells were observed using confocal microscopy (Eclipse C2 Plus, Nikon).

### 2.8. RNA Isolation and Real-Time PCR

To confirm the effect of TQC on antioxidant and mitochondrial function, we analyzed the expression of antioxidant and mitochondrial biogenesis-related genes in LPS-activated RAW 264.7 macrophages using real-time PCR. In brief, the cells were homogenized with the taco™ prep bead beater system (Taco, Taichung, Taiwan), and the total RNA was isolated using the RNeasy mini kit (Qiagen, Hilden, Germany). cDNA was synthesized using random hexamer primers and Accupower RT premix (Bioneer, Daejeon, Korea). All primer pairs were designed using the UCSC Genome Bioinformatics and the NCBI database (Primer-Blast, National Institutes of Health) and are listed in Table 1. Real-time PCR was performed using a SYBR green supermix (Bio-Rad, Hercules, CA, USA) on a CFX connect Real-Time PCR Detection System (Bio-Rad). Each real-time PCR was performed at least in triplicate. The expression of each target gene was normalized to β-actin expression and was expressed as the fold change relative to the blank groups.

### 2.9. Genomic DNA Preparation and Dot Blot

Genomic DNA was extracted in each group and isolated using the DNeasy Blood & Tissue Kit (Qiagen) for the quantification of the 8-OHdG amount within genomic DNA. The purified genomic DNA samples were spotted on the nitrocellulose membrane (0.2 µm pore size). The DNA was immobilized to the membrane by baking at 80 °C for 2 h. The membrane was then blocked with 5% skim milk and incubated with mouse anti-dsDNA (1:2000, Abcam) and mouse anti-8-OHdG (1:200, Santa Cruz) at RT overnight. Horseradish peroxidase-conjugated anti-mouse antibody (1:1000, Abcam) was incubated for 1 h at RT and visualized by enhanced Clarity Max Western ECL Substrate (ECL, Bio-rad). The relative amounts of 8-OHdG in the genomic DNA were calculated using Amersham™ Imager 600 (GE Healthcare Life Sciences, Little Chalfont, UK).

### 2.10. Western Blot

Total proteins were extracted in each group using RIPA buffer (CellNest, Minato, Tokoy, Japan) with Protease Inhibitor Cocktail Set III (1:1000, Millipore, Billerica, MA, USA). Protein concentration was measured using the Pierce™ BCA Protein Assay Kit (Thermo Fisher Scientific), according to the manufacturer’s protocol. Protein samples were separated by SDS-PAGE, transferred to a polyvinylidene difluoride (PVDF) membrane, blocked with 5% skim milk in Tris-buffered saline (TBS), and probed using various antibodies. The Western blots were visualized using ECL (Bio-rad) and exposed to Amersham™ Imager 600 (GE Healthcare Life Sciences, Uppsala, Sweden). The equivalence of protein loading was verified by probing for actin. The antibodies used were as follows: mouse anti-Mfn2 (1:500, Abcam); mouse anti-HO-1 (1:500, Abcam); rabbit anti-PGC1α (1:1000, Abcam); rabbit anti-iNOS (1:1000; Cell Signaling); rabbit anti-Nrf2 (1:1000, Abcam); rabbit anti-pink1 (1:500, Novus Biologicals, CO, USA); mouse anti-parkin (1:1000, Cell Signaling); mouse anti-β-actin (1:1000, Santa Cruz); horseradish peroxidase-conjugated anti-rabbit or -mouse antibodies (1:2500, Abcam).

### 2.11. Flow Cytometry

The MitoSOX-based flow cytometric assay was used to detect mitochondrial ROS in each group. We performed flow cytometry, as described previously [19]. First, we prepared a 0.5 mM stock solution by dissolving 50 μg of MitoSOX in 13 μL DMSO solution, and aliquoted 1.5 × 10^6^ cells into individual sterile tubes (1.5 mL) to a final volume of 0.5 mL DPBS. The cells were incubated with 1 μL DMSO as a control for background fluorescence, and all of the other samples were stained with 5 µM MitoSOX for 30 min in a CO_2_ incubator. After 30 min, the cells were washed twice with pre-warmed DPBS and analyzed using an Accuri C6 plus flow cytometer (BD bioscience). Annexin V-FITC (Abcam) was analyzed for measuring apoptosis, according to the manufacturer’s protocol.

### 2.12. Statistics

All numeric data are reported as means ± standard errors, and SPSS 18 (SPSS Inc., Chicago, IL, USA) was used for the analysis. We evaluated statistical differences using the Mann–Whitney U test. *p* values less than 0.05 were considered statistically significant.

## 3. Results

### 3.1. TQC Reduces Mitochondrial ROS and Nitric Oxide Production

To examine the effects of TQC on antioxidants in LPS-activated RAW 264.7 macrophages 24 h after treatment, we evaluated the production of NO using the Griess assay. Our results showed that the TQC extract does not induce the production of nitrogen oxides in non-activated macrophages, and these values show no significant difference from the blank without any treatment, as shown in Figure 1A. Cell viability is also not reduced in TQC treatment only, as shown in Figure 1B. However, NO levels were significantly and dose-dependently decreased following TQC extract pretreatment in LPS-activated macrophages, as shown in Figure 1A. Additionally, we confirmed the effect of the TQC extract on the inhibition of cell proliferation by cell viability assay. The cell viability rate was nearly maintained at 100% after pretreatment with the TQC extract in LPS-activated macrophages, as shown in Figure 1B. Next, we examined antioxidant activity based on hydrogen peroxide scavenging. The results of this assay showed that the TQC extract significantly inhibits hydroxyl radicals generated by H_2_O_2_, as shown in Figure 1C. We assessed the effect of TQC as an antioxidant by analyzing the change in ROS expression from the mitochondria. MitoSOX is a novel fluorogenic marker for the highly selective detection of superoxide (O_2_^−^) in the mitochondria of live cells. The MitoSOX expression was significantly reduced in the TQC groups compared with the control group in a dose-dependent manner, as shown in Figure 1D,E. Additionally, we confirmed that MitoSOX is not completely expressed upon TQC treatment in non-activated macrophages without LPS treatment, as shown in Appendix A. In contrast, LPS strongly induces the expression of MitoSOX in activated macrophages. Thus, we used the LPS-only treated macrophages as the control group for investigating the antioxidant effect of the TQC extract in LPS-activated macrophages. We analyzed the mRNA levels of TNFα, IL-6, and IL-1β; GAPDH was used as an internal control for monitoring LPS activation in RAW 264.7 macrophages, as shown in Appendix A. Compared with that in the blank group, in the LPS group, the levels of inflammation-related genes, including TNFα, IL-6, and IL-1β, significantly increased 24 h after LPS treatment. MitoSOX was quantified by flow cytometry, as shown in Figure 1F. Our data show that the TQC-treated groups gradually expressed lower percentages of positivity for MitoSOX in a dose-dependent manner, whereas the control group expressed a higher positivity for MitoSOX, as shown in Figure 1G. We showed, for the first time, that TQC effectively reduces the expression of mitochondrial ROS (MitoSOX); therefore, the TQC extract can be considered a mitochondria-targeting antioxidant.

### 3.2. TQC Inhibits the Oxidation of DNA in LPS-Activated RAW 264.7 Macrophages

We further analyzed whether the oxidative damage to DNA can be mediated by the TQC extract in LPS-activated RAW 264.7 macrophages. After pretreatment with TQC, the cells were stained with 8-OHdG to selectively stain with oxidized DNA. The 8-OHdG immunostaining revealed that the positive cells were significantly decreased in TQC-treated groups compared with those in the control group, as shown in Figure 2A. A quantification of 8-OHdG positive cells showed a substantial decrease in the intensity after treatment with up to 200 µg/mL of TQC extract, as shown in Figure 2B. To accurately determine whether the TQC extract affects the oxidation of DNA, oxidized DNA within genomic DNA, separated by groups, was confirmed by DNA dot blot analysis, as shown in Figure 2C. In the isolated genomic DNA, the amount of 8OHdG, a marker that specifically stains only oxidized DNA, was significantly reduced in the TQC-treated groups, as shown in Figure 2D. These results confirm that the TQC extract reduces oxidative DNA damage in LPS-activated RAW 264.7 macrophages.

### 3.3. TQC Induces Morphological Changes and Alterations in Mitochondria

Mitochondria dynamically change their morphology between fragmented and elongated form via mitochondrial fusion and fission. The regulation of the mitochondrial morphology is very important for the smooth supply of energy to the cells. We observed altered morphology by the labeling of mitochondria in live cells with the MitoTracker dye, as shown in Figure 3A. The mitochondrial morphology was altered to a highly fragmented form after LPS treatment, and no major difference was observed in the TQC groups. We further analyzed the expression levels of genes and proteins associated with mitochondrial morphological changes. The quantitative and qualitative control of mitochondrial morphogenetic proteins is necessary to balance the mitochondrial fission and fusion process. Dynamin-related protein 1 (*Drp1*) and mitochondrial fission 1 (*Fis1*) are associated with the promotion of mitochondrial fission, while mitofusion1 (*Mfn1*) and *Mfn2* are key factors related to mitochondrial fusion. Our data show that the *Drp1* expression level was significantly decreased after LPS treatment in the RAW 264.7 macrophages, but there was no difference in the expression level in the TQC groups and control group, as shown in Figure 3B. In contrast, the expression level of *Fis1* was significantly decreased after LPS treatment, compared with the blank group, and the expression level was significantly increased at TQC 200 µg/mL after 6 h of LPS treatment. In addition, analysis of the difference in expression level at 24 h showed that the *Fis1* expression levels in the 50 µg/mL TQC concentration was significantly increased compared to the control group, as shown in Figure 3C.

Therefore, the expression levels of *Drp1* and *Fis1*, which are key genes involved in mitochondrial fission, were significantly decreased after LPS treatment, and TQC pretreatment showed a tendency to increase the expression levels of these genes to the level of the blank group in a dose-dependent manner. This finding confirmed that the LPS treatment of RAW 264.7 macrophages led to morphological changes in the mitochondria and showed that TQC pretreatment led to changes that were indicative of recovery to the level of the blank group in a dose-dependent manner.

*Mfn1* and *Mfn2* are key genes involved in mitochondrial fusion, and their expression levels were significantly decreased at 6 h after LPS treatment. Although the TQC pretreatment of different concentrations increased the expressions of these genes on average, these increases were not significant compared with those in the control group, as shown in Figure 3D. In particular, no difference in the expression level of *Mfn2* was found between the blank and control groups at 24 h after LPS treatment, and dose-dependent changes were also not observed after TQC pretreatment, as shown in Figure 3E. However, similar to the results of real-time PCR, protein analysis by Western blotting showed that the expression level of *Mfn2* decreased at 6 h after LPS treatment and increased after TQC pretreatment in a dose-dependent manner, as shown in Figure 3F. This regulation of mitochondrial morphology is important in maintaining cellular homeostasis and normal mitochondrial function, and a proper balance between the fission and fusion processes must be maintained for the sufficient supply of energy in the cells. We confirmed that TQC induced changes in the expression levels of genes associated with changes in morphology in macrophages with LPS-induced morphological changes.

### 3.4. TQC Enhances Mitochondrial Biogenesis in LPS-Activated RAW 264.7 Macrophages

Mitochondria are important cellular organelles responsible for cellular respiration because they synthesize the ATP necessary for everyday life. We analyzed intracellular ATP levels to determine if TQC is effective in the mitochondrial functional recovery of LPS-activated macrophages. After LPS treatment, the capacity for ATP production, which represents the mitochondrial function of macrophages, significantly decreased compared with blank group. After TQC pretreatment, ATP production increased in a concentration-dependent manner and mitochondrial function was restored to the level of the blank group, but after treatment with 50 and 100 µg/mL TQC, there was no significant difference in ATP production compared with the control group. However, after treatment with 200 µg/mL of TQC, ATP production was significantly increased compared with that of the control group, and the mitochondrial function was restored to the point where the ATP level was similar to that of the blank group. Therefore, ATP production, which was reduced by LPS, was recovered to the level of the blank group after TQC pretreatment, and the increased mitochondrial ROS levels were reduced to those of the blank group, as shown in Figure 4A,B. HO-1 is an antioxidant enzyme that is an important regulator of angiogenesis, mitochondria biogenesis, and neurogenesis. In a previous study, it was reported that HO-1 overexpression enhances the biogenesis of mitochondria by increasing the protein expression of NRF1, PGC1α, and TFAM [20], and mitochondrial HO-1 is known to play important roles in regulating total protein turnover in mitochondria and in protecting against diseases, such as hypoxia, neurodegenerative diseases, or sepsis, in which substantially increased mitochondrial ROS generation has been implicated [21]. We first performed immunostaining to confirm the location of HO-1 in RAW 264.7 macrophages, as shown in Appendix A. Previous studies have shown that HO-1 localizes to several compartments within the cell, including the mitochondria [22]. Double staining with the mitochondria-specific marker MitoTracker indicated that HO-1 localized to the cytoplasm of macrophages. Hence, it can be stated that HO-1 is found exclusively in the cytoplasm of RAW 264.7 macrophages and that it localizes to the mitochondria. We also performed immunochemical staining to confirm whether TQC is also involved in the changes in the HO-1 expression level. After LPS treatment, the proportion of HO-1 positive cells did not increase compared with that in the blank group; however, with TQC pretreatment, the density of the HO-1 positive cells increased in a concentration-dependent manner, as shown in Figure 4C. Quantitatively, the HO-1 intensity in the TQC group was significantly increased in a concentration-dependent manner compared to the control group, as shown in Figure 4D. mRNA levels analyzed by real-time PCR also confirmed that the expression level of HO-1 was significantly increased in the TQC group. At a concentration of 200 µg/mL, the expression level increased by approximately 20 times or more compared to the control group, as shown in Figure 4E. However, when the mRNA expression levels of PGC1α, NRF1, and TFAM, which are mitochondrial biogenesis-related transcription factors, were measured, no differences in the expression levels were observed between the groups over time, as shown in Figure 4F–H. In addition, the protein level of HO-1 showed a significant dose-dependent increase in the TQC groups compared with control group, as shown in Figure 4I and Appendix A. Therefore, these findings revealed that TQC pretreatment resulted in a significant increase in HO-1 gene and protein expressions in the LPS-activated macrophage.

### 3.5. TQC Reduces Oxidative Stress and Mitochondrial Apoptosis in LPS-Activated RAW 264.7 Macrophages

We analyzed whether TQC inhibits the oxidation of mitochondrial DNA and regulates mitochondrial apoptosis, ultimately reducing oxidative stress. Changes in the expression level of iNOS, one of the key factors involved in the induction of oxidative stress, were confirmed by immunochemical staining, as shown in Figure 5A. At 24 h after the LPS treatment, the number and intensity of iNOS positive cells were significantly increased compared with those of the blank group, whereas the number and intensity of iNOS positive cells were significantly reduced in a dose-dependent manner in the TQC-treated group compared with the control group, as shown in Figure 5B,C. In addition, real-time PCR and Western blot showed changes in the expression levels of iNOS with respect to mRNA and protein levels. The mRNA expression levels of iNOS were also significantly reduced at 6 h and 24 h after TQC pretreatment in a dose-dependent manner compared with the control group, as shown in Figure 5D. Analysis at the protein level using Western blot also showed that the expression level of iNOS was reduced in TQC groups, as shown in Figure 5F. The quantification of protein band densitometry demonstrated that the iNOS level revealed a significant reduction in the TQC groups compared with the control group, dose-dependently, as shown in Appendix A. Nrf2 is a key transcription factor that regulates antioxidant defense and plays an important role in maintaining cellular homeostasis. We analyzed the changes in the Nrf2 expression levels and confirmed the antioxidant effect of TQC. At 6 h and 24 h after the LPS treatment of RAW 264.7 macrophages, the mRNA levels were significantly reduced compared with the blank group. However, the mRNA levels of Nrf2 for different concentrations of TQC pretreatment were significantly increased compared with the control group, as shown in Figure 5E. In addition, Western blot results showed similar increases in the expression levels of the Nrf2 protein with increasing TQC concentration, as shown in Figure 5F. The quantification of the Nrf2 protein level exhibits a significant increase in the TQC groups compared with the control group, dose-dependently, as shown in Appendix A. We found a higher inhibition of iNOS, a key factor in the induction of an oxidative environment, and expression of the antioxidant gene *NFE2L2* with an increasing concentration of TQC and, therefore, confirmed the high antioxidant activity of TQC. Moreover, the release of cytochrome c (cyt c) from the mitochondria is considered an important initial step in mitochondrial apoptosis. We first performed immunostaining to confirm the location of cytochrome c in RAW 264.7 macrophages, as shown in Appendix A. Previous studies have shown that cytochrome c localizes to the mitochondrial intermembrane/intercristal space [23]. Cytochrome c was also observed in the cytoplasm of macrophages and completely double-stained using MitoTracker. Although the exact mechanisms regulating this event are not clear, we confirmed that TQC inhibited the release of cytochrome c. Immunostaining images showed a considerable increase in the expression levels of cytochrome c following LPS treatment, and the expression levels greatly decreased after TQC pretreatment in a dose-dependent manner compared with the control group, as shown in Figure 5G. Quantitatively, the assessment of cytochrome c^+^ intensity also showed that the intensity increased significantly after LPS treatment and gradually decreased after TQC pretreatment in a dose-dependent manner, as shown in Figure 5H. Therefore, we found that the LPS-induced expression of iNOS was also significantly inhibited by TQC treatment in a concentration-dependent manner. Further, TQC enhanced Nrf2 expression in the RNA and protein levels for antioxidant defense in a dose-dependent manner. We also confirmed that TQC inhibits the activity of cytochrome c, which is released from the mitochondria and is, therefore, involved in the regulation of cell death.

### 3.6. TQC Prevents Apoptosis by Inhibiting BAX/Caspase-3 Expression and Induces Parkin Expression in LPS-Activated Macrophages

Previous studies have shown that HO-1/Nrf2 modulates autophagy and inhibits apoptosis [24,25,26]. We additionally confirmed whether TQC inhibits oxidative stress-induced apoptosis and regulates mitophagy via the Pink1–Parkin signaling pathway to better understand the underlying mechanism. We examined apoptotic cells using flow cytometry with annexin V. These data show that TQC inhibits the uptake of apoptotic cells by binding to annexin V in a dose dependent manner, as shown in Appendix A. In addition, studies on the mechanism underlying apoptosis have shown that BAX induces apoptosis with early cyto c release and caspase 3 activation [27,28]. We examined BAX and Caspase 3 expression levels using immunocytochemical staining, as shown in Figure 6A. BAX and Caspase 3 expression levels were significantly decreased after TQC treatment in a dose-dependent manner, as shown in Figure 6B,C. Moreover, PTEN-induced putative kinase 1 (Pink1) and Parkin are the most well-documented mitophagy signaling pathway mediators [29,30]. Mutations in PINK1 and Parkin cause mitochondrial dysfunction and recessive Parkinson’s disease [31,32]. Particularly, parkin is recruited selectively to dysfunctional mitochondria and promotes their autophagy [33]. We performed a Western blot assay to examine Pink1 and Parkin expression, as shown in Figure 6D. Results of the quantification of the relative protein expression level of pink1 did not significantly differ between the groups; however, the Parkin expression levels significantly decreased in the control group and increased at 200 µg/mL TQC compared with the control group, as shown in Figure 6E. TQC was found to suppress apoptosis via the BAX and caspase-3 pathways and induce Parkin expression in the LPS-activated macrophages.

## 4. Discussion

Oxidative stress plays an important role in the pathogenesis of metabolic syndrome, rheumatoid arthritis, and neurodegenerative disease [34,35]. Studies examining the efficacy of antioxidants and the association between ROS and diseases have been continuously published. Recent studies on the mechanisms of oxidative stress have reported that the pro-apoptotic protein release from the mitochondrial intermembrane space and the excessive production of ROS induce cell death [36,37]. However, only a few studies have investigated the exact mechanism by which ROS causes cellular dysfunction and the dose-dependent metabolic processes and underlying mechanisms of the formation of antioxidants from various materials, including plant leaves, fruits, and extracts from plants. We confirmed the increased release of ROS produced in the mitochondria, oxidative DNA damage, and changes in apoptosis regulation in LPS-activated macrophages. In other words, the release of cytochrome c and changes in mitochondrial function were found to play important roles in the cells, and ROS generation was found to induce cell death. A large number of studies have already been published on the role of the mitochondria in the regulation of cell death [38,39]. The present study is the first to confirm the association of TQC, which was already known as an excellent natural antioxidant substance, with the roles of mitochondria in oxidation regulation and ROS metabolism. TQC has a long history of medicinal use in home remedies, including its use as an antibiotic, antitussive agent, anti-flatulent agent, and in the treatment of nine worms (internal worms and parasites), indigestion, gastroenteritis, and bronchitis. It is also widely used in the food and perfume industries due to its distinctive smell [40,41]. It is important to establish the objective and scientific evidence of these various effects of TQC through studies investigating the mechanisms and safety of these effects. We confirmed that TQC restores reduced mitochondrial function, which was induced by LPS, and facilitates the sufficient supply of oxygen needed for energy metabolism, thereby aiding in ATP synthesis. We also confirmed that TQC inhibits the LPS-induced increase in oxidative factor activity, which demonstrates the effects of TQC on the improvement of the oxidative stress environment. Interestingly, we confirmed the changes in the key regulators of mitochondrial morphology after LPS treatment. Mitochondria are dynamic organelles that maintain cell homeostasis by regulating their morphology. In recent years, a large number of studies have reported that mitochondrial dysfunction and morphological changes are closely associated with numerous diseases [42]. Therefore, the regulation of mitochondrial morphology is critical for sufficient energy supply in cells, and the quantitative and qualitative control of mitochondrial morphology regulating proteins is required for the optimal balance between mitochondrial fission and fusion [43,44]. Drp1 and Fis1 are involved in the facilitation of mitochondrial fission, whereas MFN1 and MFN2 are involved in mitochondrial fusion. Although immunochemical staining did not reveal clear differences in the intracellular expression levels of the two key factors of fission and fusion, DRP1 and Mfn1, real-time PCR showed clear differences in the mRNA levels at 24 h compared with those at 6 h. After LPS treatment, the expression levels of all factors involved in fission and fusion were decreased compared with those in the blank group. After TQC pretreatment, factors involved in fission and fusion showed recovery, similar to the blank group, and we confirmed strong HO-1 activity with TQC. Most interestingly, however, we also showed Nrf2 upregulation at the mRNA and protein levels. Previous studies revealed that HO-1, acting through the Nrf2 transcription factor, is linked to the activation of mitochondrial biogenesis [25,45,46]. Thus, TQC effectively enhances mitochondrial biogenesis via the activation of the HO-1/Nrf2 signaling pathway in the LPS-activated macrophages. However, there is a lack of understanding of the exact molecular mechanism of action of morphological and functional changes in mitochondria, as well as a lack of studies investigating the important regulatory sites for energy metabolism and ROS production and how they change with disease. Based on this study, we expect that new therapeutic strategies may be developed by exploring the quantitative and qualitative changes that occur in mitochondria in various diseases and by studying the exact molecular mechanisms of action and control methods.

## 5. Conclusions

The present study is the first to confirm that the antioxidant effect of TQC is associated with the recovery of mitochondrial function. We confirmed that an increased number of oxidized mitochondrial DNA and mitochondrial dysfunction under oxidative stress were induced by LPS in macrophages. We also demonstrated the effects of TQC pretreatment on the inhibition of mitochondrial DNA oxidation and the recovery of mitochondrial function, thereby unraveling a new underlying mechanism of its antioxidant activity. In particular, the inhibitory effect of TQC on mitochondrial DNA oxidation was clearly observed in a dose-dependent manner, and the ATP synthesis rate and ATP levels, which are commonly used markers for mitochondrial function, showed that TQC was effective in improving mitochondrial dysfunction in an oxidative stress environment. Improvements in mitochondrial function and the restoration of cellular function by identifying new effects of TQC, including its inhibitory effect on mitochondrial DNA oxidation and restorative effect on mitochondrial dysfunction, may act as new therapeutic strategies for various diseases. The findings of this study will serve as an important guide for determining the direction of standard treatment in controlling the oxidative stress environment for many existing natural antioxidants in the future.

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
