# Peer review of "Antioxidative Effects of Thymus quinquecostatus CELAK through Mitochondrial Biogenesis Improvement in RAW 264.7 Macrophages"

_antioxidants, 2020, doi:10.3390/antiox9060548_

Round 1

Reviewer 1 Report

This manuscript by Hong et al. is interesting in that the authors investigate the protective effects of TQC in LPS treated macrophages. They found that TQC reduced mitochondrial ROS, mitochondrial DNA oxidation, and enhances HO-1 expression. The data is interesting and has implications in understanding the antioxidant properties associated with TQC. There are several concerns raised with the data presented, and these are summarized below.

  1. The role of mitochondrial dysfunction and TQC has been reported by several other groups, including its antiapoptotic function, altering mitochondrial membrane potential, and promoting mitochondrial function. This manuscript extends these findings, but does not provide a clear mechanism.

  1. I assume the authors treat with LPS and then treat with TQC, this is not clear… However, the authors need to include TQC controls, i.e. macrophages treated with control and TQC in all experiments.

  1. Several of the conclusions are incorrect. Line 266: Drp1 did not increase with TQC. Line 276: Mfn2 mRNA levels did not increase with TQC. Line 321: The authors do not show that OH-1 mediates mitochondrial biogenesis.

  1. Can the authors expand on the “morphological changes in mitochondria” that occurred with using mitoTracker– Line 251 and Fig. 3A?

  1. The HO-1 data is interesting, what is the localization of HO-1 and how is it mediating mitochondrial biogenesis?

  1. The authors found differences in mRNA and protein expression, what are the mechanisms for this?

  1. The data in Fig. 5G shows a reduction in cytochrome c expression. In order to determine release, the authors need to isolate mitochondrial and cytosol fractions to determine cytochrome c expression in each fraction. The release of cytochrome c into the cytosol is indicative of apoptosis, the authors need to clarify.

  1. There are several grammatical errors and misspellings. The authors repeatedly state that “TQC mediates mitochondrial ROS…” this is incorrect according to the presented data.

Author Response

Dear Reviewer

We were very pleased with the favorable reviews on our manuscript. We thank you for your thoughtful and helpful comments. We have extensively refined our presentation of the manuscript, figures, and references in accordance with your suggestions and the changes are marked in RED color in the revised manuscript. Below, please find our point-by-point responses to all raised queries.

Kind regards,

In-Hyuk Ha, M.D., Ph.D.

Author's Reply to the Review Report (Reviewer 1)

Comments and Suggestions for Authors

This manuscript by Hong et al. is interesting in that the authors investigate the protective effects of TQC in LPS treated macrophages. They found that TQC reduced mitochondrial ROS, mitochondrial DNA oxidation, and enhances HO-1 expression. The data is interesting and has implications in understanding the antioxidant properties associated with TQC. There are several concerns raised with the data presented, and these are summarized below.

  1. The role of mitochondrial dysfunction and TQC has been reported by several other groups, including its antiapoptotic function, altering mitochondrial membrane potential, and promoting mitochondrial function. This manuscript extends these findings, but does not provide a clear mechanism.

- We agree with the reviewer’s comment. We examined the impact of TQC, in which heme oxygenase-1 (HO-1) was transiently overexpressed and was associated with protection against oxidative stress. Actually, previous studies have shown that HO-1/Nrf2 attenuates apoptosis (Petrache et al., Am J Physiol Lung Cell Mol Physiol, 2000. DOI:10.1152/ajplung.2000.278.2.L312/ Wagner et al., Kidney International, 2003. DOI:10.1046/j.1523-1755.2003.00897.x/ Piantadosi et al, Circulation Research, 2008. DOI:10.1161/01.RES.0000338597.71702.ad). We additionally confirmed that whether TQC inhibits oxidative stress-induced apoptosis by regulating HO-1 expression using FACS analysis to understand the mechanism better. We have added the results in Supplementary FigureS3, as per the reviewer’s comment (Line. 445)

“Actually, previous studies have shown that HO-1/Nrf2 modulates autophagy and inhibits apoptosis [32-34]. We additionally confirmed that whether TQC inhibits oxidative stress-induced apoptosis by regulating HO-1/Nrf2 expression using FACS analysis to understand the mechanism better. The data showed that TQC inhibits the uptake of apoptotic cells by Annexin V Binding (Supplementary FigureS3).”

  1. I assume the authors treat with LPS and then treat with TQC, this is not clear… However, the authors need to include TQC controls, i.e. macrophages treated with control and TQC in all experiments.

- We apologize for the inconsistency in the description of drug treatment. In the revised manuscript, we have corrected drug treatment in detail in the methods section (Line. 78) as follows:

“After 1 h incubation of the cells with extracts, LPS was added to a final concentration of 1 μg/ml for 24 h. Samples were then divided into 5 groups (n = 6/group): Control group; no-treatment group; LPS group; LPS only treatment group; TQC_50 group; 50 µg/ml TQC pretreated + LPS group; TQC_100 group; 100 µg/ml TQC pretreated + LPS group; TQC_200 group; 200 µg/ml TQC pretreated + LPS group.”

- In this study, we attempted to check the TQC antioxidant effect in LPS-activated macrophages. As per the reviewer’s comment, we performed NO, cell viability assay, and MitoSox staining to confirm the toxicity and mitochondrial ROS generation after TQC treatment only in non-activated macrophages, as shown in Figure 1A-B (NO & Cell viability assay) and Supplementary FigureS1 (MitoSox).

  1. Several of the conclusions are incorrect. Line 266: Drp1 did not increase with TQC. Line 276: Mfn2 mRNA levels did not increase with TQC. Line 321: The authors do not show that OH-1 mediates mitochondrial biogenesis.

- We apologize for the insufficient explanation. We have corrected the description of Figure 3 on Lines 273, 289, and 341 as follows:

“Our data showed that Drp1 expression level was significantly decreased after LPS treatment in RAW cells, but there was no difference in the expression level in TQC groups and LPS group (Figure. 3B). In contrast, the expression level of Fis1 was significantly decreased after LPS treatment compared with the control group, and the expression level was significantly increased at 200 µg/ml after 6 h of TQC treatment.”

“In particular, no difference in the expression level of Mfn2 was found between the LPS and control groups at 24 h after LPS treatment, and dose-dependent changes were also not observed after TQC treatment (Figure. 3E). However, similar to the results of real-time PCR, protein analysis by western blotting showed the expression level of Mfn2 was decreased at 6 h after LPS treatment and increased after TQC treatment in a dose-dependent manner (Figure. 3F).”

“Therefore, these findings revealed that TQC treatment resulted in a significant increase of HO-1 gene and protein expression in LPS-activated macrophage.”

  1. Can the authors expand on the “morphological changes in mitochondria” that occurred with using mitoTracker– Line 251 and Fig. 3A?

- We appreciate the reviewer’s comment, and have now added this in the result section as follows (Line 263);

 “Mitochondria dynamically change their morphology between fragmented and elongated form via mitochondrial fusion and fission. The regulation of the mitochondrial morphology is very important for smooth supply of energy to cells. We observed altered morphology by labeling of mitochondria in live cells with the MitoTracker dye (Figure. 3A). The mitochondrial morphology was altered to a highly fragmented form after LPS treatment, and no major difference was observed in the TQC groups.”

  1. The HO-1 data is interesting, what is the localization of HO-1 and how is it mediating mitochondrial biogenesis?

- HO-1 can be localized in the mitochondria. Previous studies have shown that mitochondrial localization of HO-1 is a cytoprotective mechanism against mitochondrial oxidative stress and apoptosis (Bindu et al., Journal of biological chemistry, 2011. DOI: 10.1074/jbc.M111.279893). We also performed immunostaining to confirm HO-1 localization in the mitochondria of RAW 264.7 macrophages; this has been added to Supplementary FigureS2 (Line 325).

“And mitochondrial HO-1 is known to play important roles in regulating total protein turnover in mitochondria and in protecting against diseases such as hypoxia, neurodegenerative diseases, or sepsis, in which substantially increased mitochondrial ROS generation has been implicated [18]. We examined first to expression HO-1 in macrophage mitochondria (Supplementary FigureS2)”

- In addition, we identified an Nrf2/HO-1 regulatory cycle for mitochondrial biogenesis, and this mechanism of Nrf2/HO-1 action in mitochondrial biogenesis has been proved in previous studies (Piantadosi et al., Circulation Research, 2008. DOI:10.1161/01.RES.0000338597.71702.ad/ Piantadosi et al., Journal of biological chemistry, 2011. DOI: 10.1074/jbc.M110.207738/ MacGarvey et al., American Journal of Respiratory and Critical Care Medicine, 2012. DOI: 10.1164/rccm.201106-1152OC). We also added references for this mechanism and the following content in discussion section (Line 441);

“We confirmed strong HO-1 activity with TQC, but most interestingly, we also showed Nrf2 upregulation at the mRNA and protein levels. Previous studies revealed that HO-1, acting through the Nrf2 transcription factor, is linked to activation of mitochondrial biogenesis [30-32]. Thus, TQC effectively enhances mitochondrial biogenesis via activation of the HO-1/Nrf2 signaling pathway in LPS-activated macrophages.”

  1. The authors found differences in mRNA and protein expression, what are the mechanisms for this?

- We confirmed that TQC treatment resulted in a significant increase in HO-1 gene and protein expression in LPS-activated macrophages. In the case of HO-1, a similar pattern was observed for mRNA and protein levels. Additionally, we confirmed that PGC1α and Mfn2 showed similar mRNA and protein levels. We added quantification data of western blotting to confirm the difference in expression levels between mRNA and protein in Supplementary FigureS6.

  1. The data in Fig. 5G shows a reduction in cytochrome c expression. In order to determine release, the authors need to isolate mitochondrial and cytosol fractions to determine cytochrome c expression in each fraction. The release of cytochrome c into the cytosol is indicative of apoptosis, the authors need to clarify.

- We completely agree with the reviewer’s comment. The release of cytochrome c (cyto-c) from mitochondria is a critical event in apoptosis. Previous studies have shown that cyto-c co-localized with the mitochondria-specific dye, Mitotracker, thus indicating cyto-c release from the mitochondria (Okada et al., Am J Physiol Heart Circ Physiol, 2005. DOI:10.1152/ajpheart.00462.2005)

- We also performed immunostaining to determine the release of cyto-c from the mitochondria into the cytosol, and have added this to Supplementary FigureS2.

  1. There are several grammatical errors and misspellings. The authors repeatedly state that “TQC mediates mitochondrial ROS…” this is incorrect according to the presented data.

- We apologize for the grammatical errors and misspellings. We have completely corrected the text.

Reviewer 2 Report

The authors focused on the antioxidant effects of Thymus quinquecostatus CELAK (TQC) on primed RAW 264.7 macrophages.

Despite the promising results, the article must be significantly improved to be considered for publication.

Here below the authors may find major concerns in bullet points:

  1. please clearly specify the N used for every experiment;
  2. please clarify concentration used for LPS treatment;
  3. an internal control for LPS treatment should be added (ig. TNFα mRNA);
  4. all the results are based on pre-treated RAW macrophages with TQC extract followed by LPS administration. Does TQC treament have an effect also after LPS priming? In order to provide stronger evidence on TQC efficacy in dampening oxidative stress, Authors should address this point;
  5. O2- detection with MitoSox is clear. however, given the short half-life of the radical (5-10 seconds), authors should also provide levels of H2O2, a byproduct of O2which is more stable and easily detectable;
  6. 8-OHdG is broadly used as a marker for DNA oxidation. However, the authors should try to use a technique which could enrich mtDNA. Analysis based on genomic DNA could diluted mtDNA data; 
  7. TQC seems indeed to induce morphological changes in mitochondria, however one of the most common markers of mt-membrane dysfunction is the phosphorylation of Drp1. For this reason, authors should measure phosphorylation state of Drp1;
  8. Moreover, given that mitophagy has a protective effect in the mitochondrial anti-oxidant response, authors should look deeply into it (Pink1-Parkin axis), to give a more precise snapshot of the protective effects of TQC;
  9. is curious that upon 24 hours treatment with LPS alone NRF2 levels remained constant. Do the authors have an explanation?
  10. together with HO-1, authors should provide also levels of NQO1, another NRF2 target genes which is particularly important in the regulation of LPS-induced anti-oxidant response.
  11. indeed cytochrome C release is a common marker for ROS-induced cell death. however, authors should provide also more information about the "Apoptosome" formation: BAX/BAK and Caspases levels should be provided.
  12. Densitometries for the Western Blots should be added to the graphs.

Author Response

Dear Reviewer

We were very pleased with the favorable reviews on our manuscript. We thank you for your thoughtful and helpful comments. We have extensively refined our presentation of the manuscript, figures, and references in accordance with your suggestions and the changes are marked in RED color in the revised manuscript. Below, please find our point-by-point responses to all raised queries.

Kind regards,

In-Hyuk Ha, M.D., Ph.D.

Author's Reply to the Review Report (Reviewer 2)

Comments and Suggestions for Authors

The authors focused on the antioxidant effects of Thymus quinquecostatus CELAK (TQC) on primed RAW 264.7 macrophages.

Despite the promising results, the article must be significantly improved to be considered for publication.

Here below the authors may find major concerns in bullet points:

  1. please clearly specify the N used for every experiment;

- We have added the numbers of samples in each group in the figure legend of each figure, as follows:

Figure 1. A-B. n=6 per group, D-F. n=6 per group

Figure 2. B. n=4 per group, D. n=4 per group

Figure 3. B-E. n=4 at each time point, F. n=4 per group

Figure 4. A-B. n=6 per group, D. n=5 per group, E-H. n=4 at each time point, I-J. n=4 per group.

Figure 5. B-C. n=5 per group, D-F. n=4 per group, H. n=5 per group

  1. please clarify concentration used for LPS treatment;

- We have added the LPS concentration in the Methods section (Line 78) as follows:

 “After 1 h incubation of the cells with extracts, LPS was added to a final concentration of 1 μg/ml for 24 h. Samples were then divided into 5 groups (n = 6/group): Control group; no-treatment group; LPS group; LPS only treatment group; TQC_50 group; 50 µg/ml TQC pretreated + LPS group; TQC_100 group; 100 µg/ml TQC pretreated + LPS group; TQC_200 group; 200 µg/mL TQC pretreated + LPS group.”

  1. an internal control for LPS treatment should be added (ig. TNFα mRNA);

- We appreciate the reviewer’s accurate comments. We analyzed the mRNA levels of TNFα, IL-6, and IL-1β and normalized them using GAPDH expression in RAW 264.7 cells stimulated by LPS; this information has been added to Supplementary FigureS1.

  1. all the results are based on pre-treated RAW macrophages with TQC extract followed by LPS administration. Does TQC treatment have an effect also after LPS priming? In order to provide stronger evidence on TQC efficacy in dampening oxidative stress, Authors should address this point;

- We completely agree with this reviewer’s comment, many previous studies have provided clear evidence that pretreatment method can improve the extract efficiency, and extract pretreatment has been proven to be an important method for increasing their endogenous antioxidant activity against free radicals in previous studies. (Vuong et al., Immunopharmacology and Immunotoxicology, 2019. DOI:10.1080/08923973.2019.1569049/ Wang et al, antioxidants, 2019. DOI: 10.3390/antiox8080270/. Zhang et al., Immunopharmacology and Immunotoxicology, 2018. DOI:10.1080/08923973.2018.1424896 )

  1. O2- detection with MitoSox is clear. however, given the short half-life of the radical (5-10 seconds), authors should also provide levels of H2O2, a byproduct of O2- which is more stable and easily detectable;

- We highly appreciate the reviewer’s accurate comments. We performed H2O2 measurements to analyze the ROS scavenging activity of TQC in more detail in LPS-activated macrophages. This information has been added to Figure1C as follows:

“Next, we examined the antioxidant activity based on hydrogen peroxide scavenging. The results of this assay showed that TQC extract significantly inhibits hydroxyl radicals generated by the reaction H2O2 (Figure. 1C).”

  1. 8-OHdG is broadly used as a marker for DNA oxidation. However, the authors should try to use a technique which could enrich mtDNA. Analysis based on genomic DNA could diluted mtDNA data;

- We completely agree with this reviewer’s comment. The most widely employed method to measure oxidative damage to mtDNA and nDNA is the 8-OHdG method (Hayakava et al., 1992; Richter, 1995). Previous studies have proved that immunocytochemical analysis can help distinguish between nDNA and mtDNA based on the detection of 8-OHdG in situ in the nuclei for nDNA and in the cytoplasm for mtDNA damage (Cao et al., Toxicological Sciences. 2006. DOI:10.1093/toxsci/kfj153).

Our results implied the possibility that mtDNA contributes to the 8-OHdG signal because it was expressed in the cytoplasm of cells. However, we did not distinguish between nDNA and mtDNA in terms of the source of the increased 8-OHdG levels. We attempted to isolate mtDNA from macrophages using differential centrifugation (kim et al., scientific reports, 2018. DOI:10.1038/s41598-018-21539-y), as suggested by the reviewer. However, we obtained a very low yield of mtDNA (30 ng/μL). Various methods/protocols are available for mtDNA isolation from commercially available kits. We revised the result description as follows (Line 242):

“We further analyzed whether the oxidative damage to DNA can be mediated by TQC extract in LPS-induced RAW cells. After treatment with TQC, cells were stained with 8-OHdG to selectively stain with oxidized DNA. 8-OHdG immunostaining revealed that the positive cells were significantly decreased in TQC-treated groups compared with those in the LPS group (Figure 2A). A quantification of 8-OHdG positive cells showed a substantial decrease in the intensity after treatment with up to 200 µg/ml of TQC extract (Figure 2B). To accurately determine whether the TQC extract affects the oxidation of DNA, oxidized DNA within genomic DNA separated by groups was confirmed by DNA dot-blot analysis (Figure 2C). In the isolated genomic DNA, the amount of 8OHdG, a marker that specifically stains only oxidized DNA, was significantly reduced in the TQC-treated groups (Figure 2D). These results confirm that the TQC extract reduces oxidative DNA damage in LPS-activated RAW 264.7 macrophages.”

  1. TQC seems indeed to induce morphological changes in mitochondria, however one of the most common markers of mt-membrane dysfunction is the phosphorylation of Drp1. For this reason, authors should measure phosphorylation state of Drp1;

- We apologize for the incorrect explanation. Our data showed that Drp1 levels did not increase with TQC (Figure 3B). In contrast, the mRNA level of Fis1 was significantly upregulated at both 6 h and 24 h in the TQC 50 and 200 groups compared with that in the LPS group (Figure 3C).”

- Previously, we examined the expression levels of total Drp1 using western blot (data not shown). However, no significant differences were observed between groups.

  1. Moreover, given that mitophagy has a protective effect in the mitochondrial anti-oxidant response, authors should look deeply into it (Pink1-Parkin axis), to give a more precise snapshot of the protective effects of TQC;

- We performed western blotting to examine the pink1/parkin axis more clearly and have added the result to Supplementary FigureS5. We also added references for this mechanism and the following content in discussion section (Line 453) as follow;

“Moreover, we performed western blot assay to examine the PTEN-induced putative kinase 1 (PINK1) and Parkin expression. The PINK1 and Parkin are the most well-documented mitophagy signaling pathway [37, 38]. PINK1 or Parkin mutations cause mitochondrial dysfunction and recessive Parkinson's diseases [39, 40]. We found that TQC mediates mitophagy by parkin signaling pathway in LPS-activated macrophages (Supplementary FigureS5).”

  1. is curious that upon 24 hours treatment with LPS alone NRF2 levels remained constant. Do the authors have an explanation?

- Previous studies have assessed the transcriptional activity changes of Nrf2 and have shown that at 12, 24, 36, 48, 60, and 72 h after LPS/H2O2 treatment, the activity of Nrf2 was significantly decreased when compared with the control group (Zeng et al., Scientific reports, 2015. DOI: 10.1038/srep11100).

- And, under oxidative stress, NRF2 is not degraded, but instead travels to the nucleus where it binds to a DNA promoter and initiates transcription of antioxidative genes and their proteins. (Yamamoto et al., Molecular and Cellular Biology, 2008. DOI:10.1128/MCB.01704-07/ Sekhar et al., Toxicology and Applied Pharmacology, 2010 DOI:10.1016/j.taap.2009.06.016)

10 .together with HO-1, authors should provide also levels of NQO1, another NRF2 target genes which is particularly important in the regulation of LPS-induced anti-oxidant response.

- We appreciate this reviewer’s accurate comments. In this study, we found that TQC produced anti-oxidant effect mainly through activating Nrf2/HO-1 pathway. Previous mechanism studies have shown that the downstream proteins (HO-1 and NQO1) of Nrf2 pathway were investigated (Luo et al., Frontiers in Pharmacology, 2018. DOI: 10.3389/fphar.2018.00911/ Ci et al., Cell Death and Disease, 2017. DOI:10.1038/cddis.2017.39/ Kobayash et al., Nature communications, 2016. DOI: 10.1038/ncomms11624.).

  1. indeed cytochrome C release is a common marker for ROS-induced cell death. however, authors should provide also more information about the "Apoptosome" formation: BAX/BAK and Caspases levels should be provided.

- In this study, we analyzed cytochrome-c release from the mitochondria during the early stages of apoptosis, although the precise mechanisms regulating this event remain unclear. Previous studies have shown that during apoptosis, cytochrome-c is released from the mitochondria to the cytosol to activate a caspase cascade, which commits the cell to the death process. It has been proposed that the release of cytochrome c is caused by a swelling of the mitochondrial matrix triggered by apoptotic stimuli (Gao et al., Journal of Cell Science, 2001).

- We agree with the reviewer’s opinion. We examined the expression level of BAX and Caspase 3 using immunocytochemical staining. And, added a quantified graph and images in Supplementary FigureS4. We also added references and the following content in discussion section (Line 449) as follow;

“In addition, studies on the mechanism of action of apoptosis have shown that BAX induces apoptosis with an early release of cytochrome c and activation of caspase 3 [35, 36]. We examined the expression level of BAX and Caspase 3 using immunocytochemical staining (Supplementary FigureS4). BAX and Caspase 3 expression were significantly decreased after TQC treatment in a dose-dependent manner.”

  1. Densitometries for the Western Blots should be added to the graphs.

- We have added quantified data in Supplementary FigureS6.

Round 2

Reviewer 1 Report

  1. The authors have not clearly indicated that the macrophages have been treated with LPS prior to TQC treatment. These details need to be added to the text and figure legends. Moreover, all figures need adequate controls. Only figure 1 shows TQC treated macrophages without LPS. The dose of TQC needs to be clarified.
  2. Many of the supplemental figures are not described in the text. And parts are never mentioned in the manuscript at all (Supp Fig 1 B-D).
  3. Although the authors show that LPS promotes mitochondrial localization of HO-1, there are no controls and it is still not clear what TQC does to HO-1 localization…
  4. The role of cytochrome c release is still not clear, again the authors do not provide any controls nor do they include TQC in these experiments. The conclusion that “TQC inhibited the release of cytochrome c” is not substituted with any data.
  5. The role on mitophagy was added in the revised manuscript, but the flux through this pathway was not evaluated. It is unclear if “TQC mediates mitophagy by Parkin signaling …” as the authors state.
  6. The authors still repeatedly state that “TQC mediates mitochondrial ROS…” this is incorrect according to the presented data. See lines: 211, 228, 231, 387 in the revised manuscript.

Author Response

Author's Reply to the Review Report (Reviewer 1)

  1. The authors have not clearly indicated that the macrophages have been treated with LPS prior to TQC treatment. These details need to be added to the text and figure legends. Moreover, all figures need adequate controls. Only figure 1 shows TQC treated macrophages without LPS. The dose of TQC needs to be clarified.

- We apologize for not providing sufficient clarification in response to the reviewer’s comments submitted initially. Our main control group comprises macrophages treated with LPS only. This is because the mitosox results in supplemental figure S1 reveal that mitochondrial ROS levels do not increase in macrophages without LPS activation. Therefore, without LPS activation, it cannot be determined whether TQC pretreatment effectively reduces mitochondrial ROS levels.

- To determine whether the TQC extract used for pretreatment in our study effectively reduces mitochondrial ROS levels, comparison with LPS-activated macrophages is important, and in other previous studies, in which LPS treated macrophages have been used as an in vitro model of an oxidatively stressed environment, macrophages treated only with LPS were used as a control (Vuong et al., Immunopharmacology and Immunotoxicology, 2019. DOI: 10.1080/08923973.2019.1569049/ Hwang et al., International Journal of molecular medicine, 2018. DOI: 10.3892/ijmm.2018.3937/ Lee et al., Food Science and Technology, 2019. DOI: 10.1590/fst.15918/ Nguyen et al., International Journal of Molecular Sciences, 2020. DOI :10.3390/ijms21103439/ Gholijani et al., Journal of Immunotoxicology, 2016. DOI: 10.3109/1547691X.2015.1029145/ Wang et al., Antioxidants, 2019. DOI:10.3390/antiox8080270/ Tseng et al., PloS One, 2014. DOI: 10.1371/journal.pone.0086557/ Choi et al., BMC Complementary and Alternative Medicine, 2019. DOI: 10.1186/s12906-019-2659-5/ Sanjeewa et al., Fisheries and Aquatic Sciences, 2019. DOI: 10.1186/s41240-019-0121-8/ Zhou et al., Pharmacognosy Magazine, 2017. DOI: 10.4103/pm.pm_323_16)

- To provide more extensive background on the TQC pretreatments performed in this study, we have added relevant descriptions and related references to the introduction (Line 57).

“Many previous studies have provided clear evidence that extract pretreatment is important for improving extract efficiency and increasing their endogenous antioxidant activity against free radicals [16-18]. In the present study, we also investigated the antioxidative effects of TQC extract pretreatment associated with changes in mitochondrial function for 1 h before a 24-h treatment with LPS.”

- Additionally, the fact that the TQC extract itself is not toxic is indicated (Figures 1a and 1b). The TQC extract does not induce production of nitrogen oxides in non-activated macrophages—these values show no significant difference from the blank without any treatment. Cell viability is also not reduced (Line 213).

- TQC concentrations are indicated in all main figures, and the TQC concentration that had not been specified in supplementary data has now been inserted.

- We have outlined our experimental procedures in more detail in a timetable, which we have added to the methods section as scheme 1 (Line 89).

Scheme 1. Scheme of experimental procedure. RAW 264.7 macrophages were seeded in cell culture plate, and then pre-treated with TQC extract at various concentration (50, 100 and 200μg/ml). After 1 h of cells with TQC, LPS was added to a final concentration of 1 μg/ml for 6 or 24 h. Samples are used to analyze the antioxidant effect of TQC in LPS-activated macrophage.”

- Additionally, group names in figures and throughout the manuscript have been modified to more accurately identify control groups.

  • Samples were then divided into 5 groups (n = 6/group):
  • Blank group; no-treatment group;
  • Control group; LPS only treatment group;
  • TQC_50 group; 50 µg/ml TQC pretreated + LPS group;
  • TQC_100 group; 100 µg/ml TQC pretreated + LPS group;
  • TQC_200 group; 200 µg/ml TQC pretreated + LPS group.
  •  

  1. Many of the supplemental figures are not described in the text. And parts are never mentioned in the manuscript at all (Supp Fig 1 B-D).

- Thank you for highlighting these misses and apologize for the same. We have now made the following changes in the revised manuscript.

1) We have cited supplementary figure S1 in line 227 as follows:

“Additionally, we confirmed that MitoSOX is not completely expressed upon TQC treatment in non-activated macrophages without LPS treatment (Supplementary FigureS1A). In contrast, LPS strongly induces the expression of MitoSox in activated macrophages. Thus, we used the LPS only treated macrophages as the control group for investigating the antioxidant effect of TQC extract in LPS-activated macrophages. We analyzed the mRNA levels of TNFα, IL-6, and IL-1β; GAPDH was used as an internal control for monitoring LPS activation in RAW 264.7 macrophages (Supplementary figureS1B-D). Compared with that in the blank group, in the LPS group, the levels of inflammation-related genes, including TNFα, IL-6, and IL-1β, were significantly increased at 24 h after LPS treatment.

2) We have cited supplementary figure S2 in line 339 and 398 as follows:

“We first performed immunostaining to confirm the location of HO-1 in RAW 264.7 macrophages (Supplementary figureS2A). Previous studies have shown that HO-1 localizes to several compartments within the cell, including the mitochondria [22]. Double staining with the mitochondria-specific marker MitoTracker indicated that HO-1 localized to the cytoplasm of macrophages. Hence, it can be stated that HO-1 is found exclusively in the cytoplasm of RAW 264.7 macrophages and that it localizes to the mitochondria.”

“We first performed immunostaining to confirm the location of cytochrome c in RAW 264.7 macrophages (Supplementary FigureS2B). Previous studies have shown that cytochrome c localizes to the mitochondrial intermembrane/intercristal space [23]. cytochrome c was also observed in the cytoplasm of macrophages and completely double-stained using MitoTracker.”

3) We have cited supplementary figure S3 in line 431 as follows:

“We examined apoptotic cells using flow cytometry with annexin V. We found that TQC inhibits the uptake of apoptotic cells by binding to annexin V in a dose dependent manner.”

4) We have cited supplementary figure S4 in line 355, 384 and 393 as follows:

“In addition, the protein level of HO-1 showed a significant dose-dependent increase in TQC groups compared with control group.”

“Quantification of protein bands densitometry demonstrated that iNOS level revealed significant reduction at TQC groups compared with control group in dose-dependently.”

“The quantification of Nrf2 protein level exhibits significant increase at TQC groups compared with control group in dose-dependently.”

  1. Although the authors show that LPS promotes mitochondrial localization of HO-1, there are no controls and it is still not clear what TQC does to HO-1 localization…

- We completely agree with this reviewer’s comment, LPS does not promote mitochondrial localization of HO-1.

Compared with macrophages treated with LPS only, we confirmed an increase in HO-1 expression with TQC pretreatment.

In our analyses of HO-1-related immunostained images, RNA and protein levels, the group treated with only LPS did not show significantly different expression compared to the untreated group (blank group).

However, we have confirmed dose-dependent increases in HO-1 expression in the LPS treated group pretreated with TQC.

Additionally, although it has already been demonstrated in other references, we performed double staining with mitotracker (a specific marker of mitochondria) and HO-1 to confirm whether HO-1 was secreted from the mitochondria.

As shown in the supplementary figureS2, images of cells co-stained with HO-1 and mitotracker can be readily identified.

Therefore, we have confirmed that TQC increases HO-1 expression, and this increase is related to mitochondrial biogenesis. This is because HO-1, a representative factor reported in many prior studies to be involved in recovery after mitochondrial dysfunction via its anti-apoptotic effect under mitochondrial oxidative stress, is released from the mitochondria.

  1. The role of cytochrome c release is still not clear, again the authors do not provide any controls nor do they include TQC in these experiments. The conclusion that “TQC inhibited the release of cytochrome c” is not substituted with any data.

- We completely agree with your comment. In our present study, we performed only immunocytochemical analysis using cyt c antibody. Although the precise underlying mechanisms remain unclear, we found that TQC inhibits the expression of cyt c in a dose-dependent manner when compared with LPS-only treated macrophages as the control group. Previous studies have shown that cyt c is released from the mitochondria during the early stages of apoptosis, activating a caspase cascade, committing the cell to apoptosis. Further, it has been proposed that cyt c is released due to swelling of the mitochondrial matrix, which is triggered by apoptotic stimuli (Gao et al., Journal of Cell Science, 2001). However, our data are still insufficient to support this fact as per your comment.

- We additionally evaluated BAX and caspase levels using immunostaining to obtain more information regarding mitochondrial apoptosis. Bax is reportedly a pro-apoptotic protein and a key regulator of the mitochondrial apoptotic pathway (Cosentino et al., Trends Cell Biol, 2018 DOI: 10.1016/j.tcb.2016.11.004). In addition, it triggers the formation of a hole at the mitochondrial membrane and promotes the release of pro-apoptotic factors (Shimizu et al., Nature, 1999. DOI: 10.1038/20959. Martinou et al., Nat Rev Mol Cell Biol, 2001. DOI: 10.1038/35048069). Bax is also considered pivotal in inducing cyt c release from the mitochondria during apoptosis (Lartigue et al., Journal of Cell Science, 2008. DOI: 10.1242/jcs.029587. Jurgensmeier et al., Proc Natl Acad Sci U S A, 1998. DOI: 10.1073/pnas.95.9.4997. Zhang et al., Scientific Reports, 2017. DOI:/10.1038/s41598-017-02825-7).

- Moreover, Bax has been shown to induce cyt c release and activate caspase, particularly caspases-3. (Finucane et al., The journal of biological chemistry, 1999). Caspase-3 is activated in apoptotic cells via both extrinsic (death ligands) and intrinsic (mitochondrial) pathways (Salvesen et al., Cell Death & Differentiation, 2002. DOI:10.1038/sj.cdd.4400963. Ghavami et al., Journal of Medical Genetics, 2009. DOI:10.1136/jmg.2009.066944). We have accordingly added immunostaining images and the results of data quantification using Bax and caspase 3 antibodies in Figure 6 (Line 426). We have also rectified our study conclusion as depicted in Figure 5 as follows (Line 412):

“We also confirmed that TQC inhibits the activity of cyt c, which is released from the mitochondria, and is therefore involved in the regulation of cell death.”

  1. The role on mitophagy was added in the revised manuscript, but the flux through this pathway was not evaluated. It is unclear if “TQC mediates mitophagy by Parkin signaling …” as the authors state.

- We completely agree with your comment. Actually, considering mitophagy plays a protective role in the mitochondrial antioxidant response, we tried to analyze this aspect further (Pink1–Parkin axis) to present more precise snapshot of the protective effects of TQC. However, we did not explain this pathway in detail and evaluate it clearly, as rightly indicated by you. We have added more, relevant references and have described the results in Figure 6 (Line 437) as follows:

Moreover, PTEN-induced putative kinase 1 (Pink1) and Parkin are the most well-documented mitophagy signaling pathway mediators [29, 30]. Mutations in PINK1 and Parkin cause mitochondrial dysfunction and recessive Parkinson's disease [31, 32]. Particularly, parkin is recruited selectively to dysfunctional mitochondria and promotes their autophagy [33]. We performed western blot assay to examine the Pink1 and Parkin expression (Figure 6D). Results of the quantification of the relative protein expression level of pink1 did not significantly differ between the groups; however, Parkin expression levels were significantly decreased in the control group and increased at 200 µg/ml of TQC compared with that in the control group (Figure 6E). TQC was found to suppress apoptosis via the BAX and caspase-3 pathways and induce Parkin expression in LPS-activated macrophages.”

  1. The authors still repeatedly state that “TQC mediates mitochondrial ROS…” this is incorrect according to the presented data. See lines: 211, 228, 231, 387 in the revised manuscript.

- We apologize for the repeated error; we have now corrected the text as follows:

“TQC reduces mitochondrial ROS and nitric oxide production. (Line 211)”

“We showed for the first time that TQC effectively reduces the expression of mitochondrial ROS (MitoSOX); therefore, the TQC extract can be considered a mitochondria-targeting antioxidant. (Line 237)”

“TQC reduces mitochondrial ROS and nitric oxide production. (Line 242)”

“Therefore, we found that LPS-induced expression of iNOS was also significantly inhibited by TQC treatment in a concentration-dependent manner. Further, TQC enhanced Nrf2 expression in RNA and protein levels for antioxidant defense in a dose-dependent manner. We also confirmed that TQC inhibits the activity of cyt c, which is released from the mitochondria, and is therefore involved in the regulation of cell death. (Line 409)”

Reviewer 2 Report

The Authors significantly improved the manuscript performing a correct set of experiments.

here below would be possible to find some minor comments:

  • In response to .4 :We completely agree with this reviewer’s comment, many previous studies have provided clear evidence that pretreatment method can improve the extract efficiency, and extract pretreatment has been proven to be an important method for increasing their endogenous antioxidant activity against free radicals in previous studies. (Vuong et al., Immunopharmacology and Immunotoxicology, 2019. DOI:10.1080/08923973.2019.1569049/ Wang et al, antioxidants, 2019. DOI: 10.3390/antiox8080270/. Zhang et al., Immunopharmacology and Immunotoxicology, 2018. DOI:10.1080/08923973.2018.1424896 ):

    Authors should consider to mention these studies in the Introduction section, to give a more complete background.
  • In response to points .8 and .11: the Authors should include data from Supplementary Figure S5 and S4 in the Results section instead of the Discussion
  • in figure S1 please provide statistics for the qPCRs data.

Overall, the manuscript findings are now more consistent, increasing the strength of the study. 

Author Response

Author's Reply to the Review Report (Reviewer 2)

The Authors significantly improved the manuscript performing a correct set of experiments.

here below would be possible to find some minor comments:

In response to .4 :We completely agree with this reviewer’s comment, many previous studies have provided clear evidence that pretreatment method can improve the extract efficiency, and extract pretreatment has been proven to be an important method for increasing their endogenous antioxidant activity against free radicals in previous studies. (Vuong et al., Immunopharmacology and Immunotoxicology, 2019. DOI:10.1080/08923973.2019.1569049/ Wang et al, antioxidants, 2019. DOI: 10.3390/antiox8080270/. Zhang et al., Immunopharmacology and Immunotoxicology, 2018. DOI:10.1080/08923973.2018.1424896 ):

Authors should consider to mention these studies in the Introduction section, to give a more complete background.

In response to points .8 and .11: the Authors should include data from Supplementary Figure S5 and S4 in the Results section instead of the Discussion

in figure S1 please provide statistics for the qPCRs data.

Overall, the manuscript findings are now more consistent, increasing the strength of the study.

- We thank you and the reviewer for your thoughtful suggestions and insights. The manuscript has benefited from these insightful suggestions. As per your comment, we have added references related to the pretreatment method and the following contents to the Introduction section (Line 57).

“Many previous studies have provided clear evidence that extract pretreatment is important for improving extract efficiency and increasing their endogenous antioxidant activity against free radicals [16-18]. In the present study, we also investigated the antioxidative effects of TQC extract pretreatment associated with changes in mitochondrial function for 1 h before a 24-h treatment with LPS.”

- We have combined supplementary figures S4 and S5 as Figure 6 in the Results section; the description and related references have been cited in Line 426.

3.6. TQC prevents apoptosis by inhibiting BAX/caspase-3 expression and induces parkin expression in LPS-activated macrophages

Actually, previous studies have shown that HO-1/Nrf2 modulates autophagy and inhibits apoptosis [24-26]. We additionally confirmed whether TQC inhibits oxidative stress-induced apoptosis and regulates mitophagy via the Pink1–Parkin signaling pathway to better understand the underlying mechanism. We examined apoptotic cells using flow cytometry with annexin V. These data showed that TQC inhibits the uptake of apoptotic cells by binding to annexin V in a dose dependent manner (Supplementary figureS3). In addition, studies on the mechanism underlying apoptosis have shown that BAX induces apoptosis with early cyto c release and caspase 3 activation [27, 28]. We examined BAX and Caspase 3 expression levels using immunocytochemical staining (Figure 6A). BAX and Caspase 3 expression levels were significantly decreased after TQC treatment in a dose-dependent manner (Figure 6B-C). Moreover, PTEN-induced putative kinase 1 (Pink1) and Parkin are the most well-documented mitophagy signaling pathway mediators [29, 30]. Mutations in PINK1 and Parkin cause mitochondrial dysfunction and recessive Parkinson's disease [31, 32]. Particularly, parkin is recruited selectively to dysfunctional mitochondria and promotes their autophagy [33]. We performed western blot assay to examine the Pink1 and Parkin expression (Figure 6D). Results of the quantification of the relative protein expression level of pink1 did not significantly differ between the groups; however, Parkin expression levels were significantly decreased in the control group and increased at 200 µg/ml of TQC compared with that in the control group (Figure 6E). TQC was found to suppress apoptosis via the BAX and caspase-3 pathways and induce Parkin expression in LPS-activated macrophages.

- We has earlier missed providing the statistics for the qPCR data in supplementary figureS1; the data have now been added in the figure legend of supplementary figureS1.

Round 3

Reviewer 1 Report

The authors addressed all concerns.